# Estimating Relative Risk When Observing Zero Events—Frequentist Inference and Bayesian Credibility Intervals

**DOI:** 10.3390/ijerph18115527

**Published:** 2021-05-21

**Authors:** Sören Möller, Linda Juel Ahrenfeldt

**Affiliations:** 1Department of Clinical Research, University of Southern Denmark, 5000 Odense C, Denmark; 2Open Patient Data Explorative Network, Odense University Hospital, 5000 Odense C, Denmark; 3Unit for Epidemiology, Biostatistics and Biodemography, Department of Public Health, University of Southern Denmark, 5000 Odense C, Denmark; lahrenfeldt@health.sdu.dk

**Keywords:** relative risk, inference, confidence intervals, credibility intervals

## Abstract

Relative risk (RR) is a preferred measure for investigating associations in clinical and epidemiological studies with dichotomous outcomes. However, if the outcome of interest is rare, it frequently occurs that no events are observed in one of the comparison groups. In this case, many of the standard methods used to obtain confidence intervals (CIs) for the RRs are not feasible, even in studies with strong statistical evidence of an association. Different strategies for solving this challenge have been suggested in the literature. This paper, which uses both mathematical arguments and statistical simulations, aims to present, compare, and discuss the different statistical approaches to obtain CIs for RRs in the case of no events in one of the comparison groups. Moreover, we compare these frequentist methods with Bayesian approaches to determine credibility intervals (CrIs) for the RRs. Our results indicate that most of the suggested approaches can be used to obtain CIs (or CrIs) for RRs in the case of no events, although one-sided intervals obtained by methods based on deliberate, probabilistic considerations should be preferred over ad hoc methods. In addition, we demonstrate that Bayesian approaches can be used to obtain CrIs in these situations. Thus, it is possible to obtain statistical inference for the RR, even in studies with no events in one of the comparison groups, and CIs for the RRs should always be provided. However, it is important to note that the obtained intervals are sensitive to the method chosen in the case of small sample sizes.

## 1. Introduction

In both observational and experimental studies, the relative risk (RR) is one of the preferred measures for reporting associations between dichotomous exposures/risk factors and a dichotomous outcome. For clinicians, the RR is often easier to interpret than an odds ratio (OR) [1]. The RR is defined as the ratio between outcome probabilities obtained in two groups and thus has a very direct probabilistic interpretation compared to the more abstract definition of the OR. Hence, while translating an RR of two to the statement “The risk in the intervention group is twice as high as in the control group”, a similar interpretation would be incorrect for an OR.

When estimating an RR, researchers may encounter a challenge if one of the comparison groups does not experience any event (or if everyone in a group experiences an event), as this would result in an RR of zero or *∞*, respectively. In these cases, confidence intervals (CIs) for the RR cannot easily be reported, and in the resulting research papers, it is often concluded that the sample was not large enough to obtain sufficient inference about the RR. However, as the RR is a measure of an association, this is not reasonable. For example, in the case of a randomized clinical trial (RCT) where the outcome is a rare adverse event, a result of no adverse event in the intervention group, but of a moderate number of adverse events in the control group, would give clear evidence of a protective effect of the intervention [2,3]. Similarly, in an epidemiological study investigating the efficacy of a vaccine, a result with no observed infections in the group of vaccinated persons, but with some infected persons among the non-vaccinated, could give clear evidence of a positive effect of the vaccine [4].

In general, it seems erroneous that a study reporting decreasing incidence of an adverse outcome from 10% to 1% can (possibly) provide strong evidence of a protective effect, while a similar sized study reporting decreasing incidence of the adverse outcome from 10% to 0% is not considered to provide strong evidence, although the effect size is larger. Hence, it is obvious that the problem with an RR estimate of zero does not indicate an epidemiological weakness of the study but rather a limitation of the statistical procedures used to obtain the estimate and corresponding CI.

This problem has been discussed by Fagerland et al. as a special case without much detail [5] and in a note on how to approach the problem in R [6]. Moreover, research [7,8] has suggested an approximate rule proposing that an upper limit for a CI could be calculated as three divided by the total number of observations. However, these suggestions were made before the development of modern computational possibilities, and they are later discussed in a more current setting [9].

The aim of this paper is to present, compare, and discuss different statistical approaches to obtain frequentist CIs as well as Bayesian credibility intervals (CrIs) in cases where an RR is estimated to be zero due to no outcome events in one of the comparison groups.

## 2. Materials and Methods

### 2.1. Motivating Examples

To demonstrate the different approaches, we will present three examples, each of which corresponds to the setting of an RCT with two groups of equal size and an observed incidence of an (adverse) outcome of 0% in the intervention group and 10% in the control group. These three examples correspond to total sample sizes of 40, 200, and 400, respectively (Table 1). They are chosen as their ranges correspond to most of the study sizes used in clinical practice. Furthermore, we include three additional examples (Examples D, E, and F), corresponding to large (*N* = 20,000) epidemiological studies with differing incidences of 10%, 1%, and 0.1%, respectively, in the reference group. We decided not to include examples in which the group with 0 events is the reference (control) group, as these result in RR estimates of *∞* and, hence, are best handled by presenting the inverse RR using the other group as the reference.

For clarity, we will denote the group experiencing 0 events as the intervention group, and the group experiencing more than 0 events as the control group, even if this nomenclature may be considered inappropriate in epidemiological studies. Moreover, we will denote positive outcomes as events and negative outcomes as non-events.

First, we will present the investigated approaches. We will start with frequentist approaches to determine CIs for the RR, continue with frequentist approaches utilizing the OR as an approximation of the RR, and end with Bayesian approaches to obtain CrI for the RRs. Subsequently, we will demonstrate the results of applying these approaches to the examples from Table 1. All computations are performed in Stata version 16.1. Stata codes are available in Appendix A.

### 2.2. Frequentist Approaches

#### 2.2.1. Normal Approximation Formula for the RR

The classical formula for a CI of the RR using the normal approximation is given by
$$\begin{array}{c} CI_{\mathrm{RR}\;N-\mathrm{appr}}=\left(\hat{RR}\cdot \exp\left(-1.96\cdot SE_{\mathrm{RR}\;N-\mathrm{appr}}\right);\hat{RR}\cdot \exp\left(1.96\cdot SE_{\mathrm{RR}\;N-\mathrm{appr}}\right)\right) \end{array}$$
with
$$\begin{array}{c} \hat{RR}=\frac{d_{1}/n_{1}}{d_{0}/n_{0}} \end{array}$$
and
$$\begin{array}{c} SE_{\mathrm{RR}\;N-\mathrm{appr}}=\sqrt{1/d_{1}-1/n_{1}+1/d_{0}-1/n_{0}} \end{array}$$
where $d_{1}$ and $d_{0}$ are the number of events in the intervention group and the control group, respectively, and $n_{1}$ and $n_{0}$ are the corresponding total number of observations in each group. Here, 1.96 is the 0.975 quantile of the normal distribution, corresponding to a 95% confidence interval. This will fail in the case of $d_{1}=0$, as it would imply division by 0 in the standard error (SE). In principle, one could try to determine limits for these expressions for $d_{1}\to 0$, handling the counts as if they were continuous. For the lower confidence limit, this will result in a limit of 0, while the (more interesting) upper limit can be shown to diverge to *∞* by use of l’Hospital’s rule. Furthermore, as the normal approximation for the CI generally requires counts of at least 5 (or 10) in each cell, this method is discouraged in case of 0 events observed in the intervention group [10]. Hence, we will not investigate modifications to the normal-based CI for the RR further.

#### 2.2.2. Adding One to Each Cell or Moving One Non-Event to Event in the Intervention Group

A simple ad-hoc approach to avoid these challenges is adding 1 to each cell in the cross table. This will result in an RR estimate slightly higher than 0 and create the possibility of applying standard methods to obtain a CI. However, because the resulting cross table will include 1 as the number of events in the intervention group, normal approximation methods are generally discouraged [10]. Furthermore, this approach will artificially increase the total sample size by 4, thus suggesting more information than what is obtained in reality, resulting in too narrow confidence intervals. In large studies, this effect will be minimal, but in small studies, the effect should be considered (Approach I in Table 2 and Table 3).

In 1966, Gart [11] suggested adding 1/2 instead of 1 to each cell in the table. This naturally decreases the distortion of the sample size. However, it has been shown [6] that the resulting CI is quite sensitive to the magnitude of the added term. Furthermore, due to the resulting counts not being integers, this solution has no clear probabilistic interpretation, while approximate methods will still be discouraged due to the low count (0.5) in one cell. Again, a one-sided CI can be considered to increase both power and interpretability. Because of the unclear probabilistic interpretation and infeasibility of applying many estimation methods due to non-integer counts, in this paper, we decided not to investigate the approach of adding 1/2 further.

A less distorting but similar strategy is to move one observation in the intervention group from no event to event. This results in an event-count of 1 and a non-event-count decreased by 1, without modifying results in the control group. This method preserves the total sample size (and the sample size in each group) as well as the integrity of all counts. One can think of this method as a counterfactual setting, investigating a similar study, where one of the intervention participants did counterfactually experience the outcome event, corresponding to an outcome with (slightly) weaker evidence for a protective effect. However, it should be noted that this will always lead to a more conservative estimate of the RR in the sense that it will indicate a weaker association than is actually present (Approach I in Table 2 and Table 3).

Moreover, as the true RR is 0, estimating a lower limit of the CI above 0 does not seem reasonable. Hence, one can obtain additional power and avoid unintuitive results by instead reporting a one-sided upper CI.

#### 2.2.3. Using the OR as Substitute for the RR

It is well known that the OR can be used as an approximation for the RR when the outcome is rare [12]. In our examples, this implies that the OR often could be a feasible approximation, as the 0 event in the intervention group indicates a low probability, while the probability in most, but not necessarily all studies, will be low in the control group as well. If necessary, the literature suggests approaches to adjust the resulting OR and its CI to achieve a closer approximation to the RR [12,13].

The classical normal approximation formula for a CI for the OR, as suggested by [14] is
$$\begin{array}{c} CI_{\mathrm{OR}\;N-\mathrm{appr}}=\left(\hat{OR}\cdot \exp\left(-1.96\cdot SE_{\mathrm{OR}\;N-\mathrm{appr}}\right);\hat{OR}\cdot \exp\left(1.96\cdot SE_{\mathrm{OR}\;N-\mathrm{appr}}\right)\right) \end{array}$$
with
$$\begin{array}{c} \hat{OR}=\frac{d_{1}/h_{1}}{d_{0}/h_{0}} \end{array}$$
and
$$\begin{array}{c} SE_{\mathrm{OR}\;N-\mathrm{appr}}=\sqrt{1/d_{1}+1/h_{1}+1/d_{0}+1/h_{0}} \end{array}$$
where $h_{1}$ and $h_{0}$ are the number of non-events in the intervention and the control group, respectively.

Similarly, as in the case of the approximate formula for RR, if we let $d_{1}\to 0$ (and $h_{1}\to n_{1}$), the lower CI limit will converge to 0, while the upper limit will diverge to *∞*. Instead, one can apply methods for combinatorically determined CIs for the OR, which typically will be able to handle counts of 0. For instance, the Cornfield approximate CI for the OR [15] can be estimated in the case of 0 events in the intervention group and will, in this case, always result in a lower CI limit of 0 (Approach II in Table 2 and Table 3).

This approach has been improved further by Baptista and Pike [16] and Fagerland [17] with exact CIs for the OR including an implementation of the exact CI suggested by Cornfield [15]. Furthermore, Fagerland [17] also provided a Stata implementation (merci), which we used to apply these methods. This implementation offers both exact CIs as well as mid-p intervals obtained by approximating a correction term [17]. As these extensions are currently computationally infeasible for large samples, we only apply these for Examples A, B, and C, with sample sizes of 40–400 and not for examples D, E, and F with a sample size of 20,000 (Approach II in Table 2).

#### 2.2.4. Frequentist Regression Models

In general, inference for the RR is possible by binomial regression (a general linear model with logarithmic link function) determining maximum likelihood estimates (MLE). In our examples, this approach will obtain an MLE at a coefficient of −∞ corresponding to an RR of 0. Hence, binomial regression will not result in meaningful estimates and CIs. As with the case of binomial regression, logistic regression is infeasible in the case of an estimated OR of 0, as the MLE will correspond to a coefficient of −∞.

#### 2.2.5. Bootstrapping

As no individuals in the intervention group experienced the outcome event, any bootstrapping sample will result in RR estimates of 0 (or possibly in non-estimability of the RR, if no events in the control group are included in the sample). Hence, a bootstrapped CI will consist of only 0 and thus be inappropriate, although it has, in some cases, been reported in recent literature [3].

### 2.3. Bayesian Approaches

#### 2.3.1. Priors for the Proportions

The most natural approach for modeling a dichotomous outcome in two groups is to assume priors for the proportion of events in both groups. The posterior of these proportions can then be estimated, and from these, the posterior of the RR can be determined.

The most natural choice of priors is a beta distribution, as this is the conjugate prior for a Bernoulli likelihood [18]. In this case, with a B(α,β) prior for a proportion, the posterior will be B(α+d,β+h) if *d* positive and *h* negative outcomes have been observed. Applying this to our examples, we have an intervention group prior of $B(\alpha_{1},\beta_{1})$, observing $d_{1}$ positive and $h_{1}$ negative outcomes, and a control group prior of $B(\alpha_{0},\beta_{0})$, observing $d_{0}$ positive and $h_{0}$ negative outcomes. In this case, the posterior for the RR will be given by
$$\begin{array}{c} P\left(RR=r\right)=\int_{p/q=r}f_{B(\alpha_{0},\beta_{0})}\left(p\right)f_{B(\alpha_{1},\beta_{1})}\left(q\right)dpdq \end{array}$$
where $f_{B(\alpha ,\beta )}$ is the density of a beta distribution with parameters α and β.

While this is challenging to evaluate algebraically, it can easily be estimated numerically, e.g., by Markov Chain Monte Carlo (MCMC) simulation. The two most natural choices for hyperparameters α and β are either B(1,1), corresponding to the uniform distribution on (0,1), or B(0.5,0.5), which is the non-informative Jeffrey’s prior for the Bernoulli model (Approach III in Table 2 and Table 3).

Furthermore, one could consider priors with a point mass (an atom) at 0, hence with a non-zero prior probability of the proportion being precisely 0. We decided not to employ these priors, as the mass placed at 0 necessarily would be a subjective choice and would strongly influence the estimated CrIs.

#### 2.3.2. Bayesian Binomial Regression

A different approach is to assume a prior for the RR itself together with a prior for the proportion in the control (reference) group, and from these two, obtain a posterior for the RR. In this case, estimation by Bayesian binomial regression is the most relevant method to apply, as binomial regression generally is the preferred strategy to obtain RR estimates from regression models [19]. An important practical point in this approach is that the Bayesian estimation cannot be initialized by MLE, as we have described above, since the MLE does not exist in our case.

If it is desired to use weakly informative normal priors, it might be necessary to use smaller standard deviations than otherwise common to enable fitting of the model in these circumstances.

## 3. Results

Results for examples A, B, and C are given in Table 2. When comparing the approaches of moving and adding one observation, respectively, we find that adding one observation generally results in narrower CIs than moving one, especially in Example A, which has the smallest sample size. As expected, the one-sided approaches result in lower upper bounds for the CI.

Comparing the approaches regarding the ORs, all CIs will automatically have zero as their lower bound. The approximate Cornfield interval generally obtains the lowest upper bound, especially for small sample sizes, but as the approximate Cornfield interval is constructed by using a large sample asymptotic theory, this interval should be interpreted carefully in small samples [15]. Among the exact approaches for OR-based CIs, the midpoint Baptista–Pike method generally results in the narrower CI, while the exact Cornfield method results in the widest intervals. For moderate sample sizes, the OR-based approaches result mainly in narrower CIs than moving or adding one.

The Bayesian proportion-based approach results in CrIs of similar magnitude compared to the frequentist OR-based CIs, with highest posterior density (HPD) intervals being generally narrower than equal-tailed (EqT) CrIs. While these CrIs have a non-zero lower limit, this estimate is extremely close to zero in all examples. On the other hand, the Bayesian approach, based on binomial regression, results in much narrower CrIs than is the case for all the other methods investigated.

Results from examples D, E, and F corresponding to larger epidemiological cohort studies are presented in Table 3. In general, these results behave similarly to those obtained from the smaller examples. Furthermore, the proportion-based Bayesian approaches result in CrIs, which are similar in magnitude to the CIs obtained by frequentist approaches. On the other hand, the Bayesian binomial regression obtains extremely narrow CrIs with almost all mass concentrated close to zero.

## 4. Discussion

In this study, results showed that several different methods are feasible for obtaining CIs or CrIs even in studies with zero observed events in the intervention group. Overall, the different frequentist approaches result in comparable CIs as long as the sample size is moderate or large, and Bayesian methods are able to obtain CrI of similar magnitude, although Bayesian binomial regression generally results in very narrow CrIs concentrating much of the mass around zero. This phenomenon is caused by the parametrization implied by binomial regression, in which an RR of zero corresponds to a regression coefficient of −∞, hence driving the posterior distribution of the coefficient downwards. Therefore, the results from Bayesian binomial regression should be interpreted cautiously and the proportion-based Bayesian approach should be considered the preferred method. Moreover, while Bayesian approaches have an explicit probabilistic interpretation, as a 95% CrI indicates an interval, which with 95% probability contains the true population parameter, conditional on the chosen prior distributions, Bayesian approaches are still not widespread in epidemiological literature [20], which limits dissemination of their results.

### 4.1. Strengths and Limitations

The main strength of this study is that both the frequentist and the Bayesian approaches, commonly suggested in the literature, are compared in a set of examples representing many practical applications, thus enabling a direct comparison of the approaches and their results.

A weakness of this study is that it had to be limited to the most well-known approaches. Thus, we were not able to compare all the approaches suggested in the literature, such as the exact approaches to the CI [21] and exact Bayesian inference [22]. Another limitation is that due to the observation of zero events laying at the border of the parameter space for all frequentist methods, no formal evaluation of coverage for the obtained CIs was feasible.

### 4.2. Possible Future Research

This paper focused on the inference for unadjusted RRs, for instance, those typically estimated in randomized clinical trials. In observational studies, one would most often have a desire to adjust for possible confounders, which would not be easily achievable with most of the discussed methods. Hence, future research should investigate how inference for RR estimates with 0 events can be obtained in models with covariate adjustment.

Furthermore, problems as those seen in the case of an RR estimate of 0 can occur in time-to-event studies, in which no events are observed in the intervention group. Similarly to the RR, this challenges the inference obtainable from, for instance, the hazard ratio from a survival analysis. This issue should be investigated in future studies.

## 5. Conclusions

This investigation demonstrates that it is possible to obtain CIs and CrIs for the RR in studies with no observed events in the intervention group. Hence, researchers should also report CIs or CrIs in these circumstances and refrain from concluding that inference was impossible due to no observed events, as this often will result in understating the evidence obtainable from a study. On the other hand, researchers should be aware that the obtained intervals are sensitive to the method applied for smaller samples while quite robust in large samples.

## Figures and Tables

**Table 1 ijerph-18-05527-t001:** Examples.

	Examples with Varying Samples Sizes		Examples with Varying Outcome Rates	
	Intervention	Control	Intervention	Control
	Example A (N=40)		Example D (N=20,000)	
Negative outcome	20	18	10,000	9000
Positive outcome	0	2	0	1000
	Example B (N=200)		Example E (N=20,000)	
Negative outcome	100	90	10,000	9900
Positive outcome	0	10	0	100
	Example C (N=400)		Example F (N=20,000)	
Negative outcome	200	180	10,000	9990
Positive outcome	0	20	0	10

**Table 2 ijerph-18-05527-t002:** Results: reported intervals are 95% CIs for frequentist methods and 95% CrI for Bayesian methods. (Note: 0 indicates an estimate formally forced to be 0, while 0.000 indicates an estimate numerically within 0.0005 of 0.)

	Example A	Example B	Example C
Approach I			
Adding 1 (two-sided)	(0.0375; 2.9625)	(0.0120; 0.6912)	(0.0065; 0.3507)
Adding 1 (one-sided)	(0; 2.0851)	(0; 0.4989)	(0; 0.2544)
Moving 1 (two-sided)	(0.0492; 5.0831)	(0.0130; 0.7666)	(0.0068; 0.3690)
Moving 1 (one-sided)	(0; 3.5011)	(0; 0.5526)	(0; 0.2676)
Approach II			
Odds ratio (approximate Cornfield)	(0; 1.8912)	(0; 0.3513)	(0; 0.1738)
Odds ratio (exact Cornfield)	(0; 5.2804)	(0; 0.4178)	(0; 0.1864)
Odds ratio (mid-p Cornfield)	(0; 3.4316)	(0; 0.3256)	(0; 0.1483)
Odds ratio (exact Baptista-Pike)	(0; 3.4316)	(0; 0.3933)	(0; 0.1870)
Odds ratio (mid-p Baptista-Pike)	(0; 2.1267)	(0; 0.3140)	(0; 0.1597)
Approach III			
Bayes B(1,1) prior EqT	(0.0072; 2.4681)	(0.0020; 0.4069)	(0.0013; 0.1657)
Bayes B(1,1) prior HPD	(0.0002; 1.8191)	(0.0002; 0.3185)	(0.0000; 0.1381)
Bayes B(0.5,0.5) prior EqT	(0.0002; 2.0744)	(0.0005; 0.2621)	(0.0001; 0.1213)
Bayes B(0.5,0.5) prior HPD	(0.0000; 1.4303)	(0.0001; 0.1901)	(0.0000; 0.0906)
Bayes binreg β∼N(0,10000) EqT	(0.0000; 0.0147)	(0.0000; 0.0047)	(0.0000; 0.0008)
Bayes binreg β∼N(0,10000) HPD	(0.0000; 0.0005)	(0.0000; 0.0002)	(0.0000; 0.0000)

**Table 3 ijerph-18-05527-t003:** Results: reported intervals are 95% CIs for frequentist methods and 95% CrI for Bayesian methods. (Note: 0 indicates an estimate formally forced to be 0, while 0.000 indicates an estimate numerically within 0.0005 of 0.)

	Example D	Example E	Example F
Approach I			
Adding 1 (two-sided)	(0.0001; 0.0071)	(0.0014; 0.0710)	(0.0117; 0.7040)
Adding 1 (one-sided)	(0; 0.0052)	(0; 0.0517)	(0; 0.5066)
Moving 1 (two-sided)	(0.0001; 0.0071)	(0.0014; 0.0717)	(0.0128; 0.7810)
Moving 1 (one-sided)	(0; 0.0052)	(0; 0.0522)	(0; 0.5612)
Approach II	(0; 0.0035)	(0; 0.0380)	(0; 0.3838)
Odds ratio (approximate Cornfield)			
Approach III			
Bayes B(1,1) prior EqT	(0.0000; 0.0038)	(0.0002; 0.0354)	(0.0036; 0.3911)
Bayes B(1,1) prior HPD	(0.0000; 0.0032)	(0.0000; 0.0292)	(0.0000; 0.3147)
Bayes B(0.5,0.5) prior EqT	(0.0000; 0.0039)	(0.0003; 0.0364)	(0.0033; 0.3730)
Bayes B(0.5,0.5) prior HPD	(0.0000; 0.0031)	(0.0000; 0.0295)	(0.0001; 0.2984)
Bayes binreg β∼N(0,10000) EqT	(0.0000; 0.00003)	(0.0000; 0.0003)	(0.0000; 0.0041)
Bayes binreg β∼N(0,10000) HPD	(0.0000; 0.00000)	(0.0000; 0.00001)	(0.0000; 0.0001)

## Data Availability

Data are contained within the article or Appendix A.

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
