# Peer review of "Estimating Relative Risk When Observing Zero Events—Frequentist Inference and Bayesian Credibility Intervals"

_ijerph, 2021, doi:10.3390/ijerph18115527_

Round 1

Reviewer 1 Report

Overall, I feel this is a great paper. The topic is extremely pertinent, as attempts to address a scenario that every investigator runs into from time to time and has difficulty on how to tackle it. In addition, the article is well written. The suggestions stated below are numerous though in general fairly minor. In general; however, I am bit concerned that the language and terminology used in the article is a little bit esoteric, limited to readers that are intimately involved in statistical analysis, especially when it comes to Bayesian methods. I feel that simplifying the article by explaining certain methods further and in more basic terms may expand the readership and the potential effectiveness of the article.

Line 1: Change “the relative risk” to just “relative risk”

Line 6: “In this paper, WHICH” doesn’t sound grammatically correct. Consider something like “This paper, which….., aims to …”

Line 18: Consider changing to observational and experimental study designs. You are leaving a lot of study designs out.

Line 22: Consider changing to “When calculating an RR, researchers may…”

Line 28: Take out comma after (RCT)

Line 43: What is this note? Where was this published? The reference format under references is not correct

Line 55: change “the (adverse) outcome” to “an (adverse) outcome” or “an adverse outcome”

Table 1: I would suggest to clarify in the as a column heading that the first set of examples represent varying sample size, while the second set represents varying incidence/rate of the outcome in the control croup

Line 64: I suggest deleting “in the following”

Line 69: comes off a bit awkward with “next section”

Line 178: I suggest changing “as the MLE…” to “since the MLE…”

Table 3: clearly state somewhere in the manuscript why all Odds ratio methods were not applied/presented for examples D, E, F

Table 2 and 3: make sure terms such as EqT, merci, midp are defined somewhere in the manuscript before their use.

Table 2 and 3: I feel that the difference between exact CI and midpoint CI need to be more clearly defined in the text.

Line 69-72: I suggest expanding this paragraph to present the reader on what to expect in the following sections describing the approaches. These sections are fairly extensive, and I feel that it would be beneficial for the reader to have a general understanding of the organization of these sections.

Table 2 and table 3: I found myself going back and forth between the tables and the previous sections describing these approaches. When initially reading these sections describing the approaches it was not always apparent the significance of certain methods in terms of how or if they are to be used in the analysis. I feel that every approach presented in the tables should be clearly highlighted, in the sections describing these approaches, that they will be used in the analysis.

Author Response

Reviewer 1

Overall, I feel this is a great paper. The topic is extremely pertinent, as attempts to address a scenario that every investigator runs into from time to time and has difficulty on how to tackle it. In addition, the article is well written. The suggestions stated below are numerous though in general fairly minor. In general; however, I am bit concerned that the language and terminology used in the article is a little bit esoteric, limited to readers that are intimately involved in statistical analysis, especially when it comes to Bayesian methods. I feel that simplifying the article by explaining certain methods further and in more basic terms may expand the readership and the potential effectiveness of the article.

ANSWER: Thank you for the positive comments. We have taken into account your suggestions (See details below) and hope the manuscript is now easier to read for a broader audience.

Line 1: Change “the relative risk” to just “relative risk”

ANSWER: Corrected (not highlighted in the tracked changes version, due to limitations in the abstract template)

Line 6: “In this paper, WHICH” doesn’t sound grammatically correct. Consider something like “This paper, which….., aims to …”

ANSWER: Corrected (not highlighted in the tracked changes version, due to limitations in the abstract template)

Line 18: Consider changing to observational and experimental study designs. You are leaving a lot of study designs out.

ANSWER: Thank you for the good suggestion, we have changed it to “observational and experimental studies”.

Line 22: Consider changing to “When calculating an RR, researchers may…”

ANSWER: This is now changed.

Line 28: Take out comma after (RCT)

ANSWER: Corrected

Line 43: What is this note? Where was this published? The reference format under references is not correct

ANSWER: The note is published online, and as far as we could determine, it has never been published in a journal. We have now corrected the reference, pointing it out as an online resource:

“6. Dewey, M.E. Collated responses from R-help on confidence intervals for risk ratios. http://www.zen103156.zen.co.uk/rr.pdf,accessed on 2021-05-12.”

Line 55: change “the (adverse) outcome” to “an (adverse) outcome” or “an adverse outcome”

ANSWER: Corrected

Table 1: I would suggest to clarify in the as a column heading that the first set of examples represent varying sample size, while the second set represents varying incidence/rate of the outcome in the control croup

ANSWER: We have now added the headings “Examples with varying samples sizes” and “Examples with varying outcome rates” to Table 1.

Line 64: I suggest deleting “in the following”

ANSWER: Corrected

Line 69: comes off a bit awkward with “next section”

ANSWER: Changed to “First,”

Line 178: I suggest changing “as the MLE…” to “since the MLE…”

ANSWER: Corrected

Table 3: clearly state somewhere in the manuscript why all Odds ratio methods were not applied/presented for examples D, E, F

ANSWER: We have now made the reason more explicit with this sentence: “As these extensions are currently computationally infeasible for large samples, we only apply these for Examples A, B, and C, with sample sizes of 40--400 and not for examples D, E, and F with sample size of 20,000.”

Table 2 and 3: make sure terms such as EqT, merci, midp are defined somewhere in the manuscript before their use.

ANSWER: We have now made explicit in the text, that “merci” is the name of Fagerland’s Stata package implementing these methods, and removed the word “merci “from Table 2 and 3. Moreover, we have added the explanation of “EqT” to the text, and corrected midp” to the more correct “mid-p” in Table 2.

Table 2 and 3: I feel that the difference between exact CI and midpoint CI need to be more clearly defined in the text.

ANSWER: We have now added an explanation of this difference to the method section: “This implementation offers both exact CIs as well as mid-p intervals obtained by approximating a correction term.”

Line 69-72: I suggest expanding this paragraph to present the reader on what to expect in the following sections describing the approaches. These sections are fairly extensive, and I feel that it would be beneficial for the reader to have a general understanding of the organization of these sections.

ANSWER: We have now expanded this paragraph with the sentence: “We will start with frequentist approaches to determine CIs for the RR, continue with frequentist approaches utilizing the OR as an approximation of the RR, and end with Bayesian approaches to obtain CrI for the RRs.”

Table 2 and table 3: I found myself going back and forth between the tables and the previous sections describing these approaches. When initially reading these sections describing the approaches it was not always apparent the significance of certain methods in terms of how or if they are to be used in the analysis. I feel that every approach presented in the tables should be clearly highlighted, in the sections describing these approaches, that they will be used in the analysis.

ANSWER: We agree that the correspondence between methods and results sections was unclear. We have now added the terms “Approach I”, “Approach II” and “Approach III” to both the subsection headings in the methods section as well as to Table 2 and 3. Furthermore, we have pointed out in the methods section which approaches that are not reported in the tables.

Reviewer 2 Report

The authors perform a comparison of a variety of frequentist and Bayesian methods for constructing confidence and credible intervals to estimate the relative risk in the case where there are zero incidences in the event group.  They test the set of methods on six examples of varying sample sizes to illustrate which are robust regardless of sample size and which must be treated with caution. 

The manuscript is well-written, but lacking in a level of description that could be helpful to someone without a strong statistical background.  As I suspect that many readers of this journal may not have extensive training in theoretical statistics, I would consider recommending this manuscript for publication once the following comments have been addressed:

  • In general, the manuscript could be strengthened by providing a bit more clarification (or at least some additional citations for follow-up reading) in several areas. Here are a few examples for which I would like to see some additional elaboration:
    1. At the beginning of the introduction, explicitly define the term “relative risk” and discuss why it is a useful metric for clinical and epidemiological studies. Comparison to the definition and interpretation for “odds ratio” would also be helpful to motivate the study.
    2. Sections 2.2.4 and 2.3.2 on binomial regression are very brief and could benefit from additional explanation or some citations for further reading. As written, these sections will be very difficult to understand by anyone without a strong statistical background.  In particular, I do not see that 2.3.2 gives the reader sufficient information to replicate the results of this study on their own.
    3. Authors should discuss the difference between frequentist confidence intervals and Bayesian credible intervals. In particular, usage of Bayesian methods is generally not as widespread as frequentist methods, and an explicit definition for a credible interval (and discussion of how it differs in interpretation from a frequentist confidence interval) should be provided.
  • Use of the “adding 1” or “moving 1” methods should be treated with extreme caution, as they bring to mind issues of data manipulation/manufacturing.  Wherever possible, it should be clarified that:
    1. Increasing the sample size through the “adding 1” method will lead to narrower confidence intervals, giving an inflated sense of precision.
    2. Use of the “moving 1” method will always lead to a more “conservative” estimate of the RR in the sense that it will indicate a weaker association than is actually present.
    3. The authors did a nice job of stating that these methods are not robust for small sample sizes in the discussion; I’d like to see a sentence to this effect added up front in the abstract as well.
  • A minor issue: please indicate that the use of z* = 1.96 in the confidence intervals in Sections 2.2.1 and 2.2.3 corresponds to a 95% CI; else, replace with the more general z* notation and indicate how these critical values may change to meet whatever confidence level you choose to impose.

Author Response

Reviewer 2

The authors perform a comparison of a variety of frequentist and Bayesian methods for constructing confidence and credible intervals to estimate the relative risk in the case where there are zero incidences in the event group.  They test the set of methods on six examples of varying sample sizes to illustrate which are robust regardless of sample size and which must be treated with caution. 

The manuscript is well-written, but lacking in a level of description that could be helpful to someone without a strong statistical background.  As I suspect that many readers of this journal may not have extensive training in theoretical statistics, I would consider recommending this manuscript for publication once the following comments have been addressed:

ANSWER: Thank you for the positive evaluation, we have now taken into account your specific comments, and hope the manuscript has become more readable to a wider epidemiological audience.

  • In general, the manuscript could be strengthened by providing a bit more clarification (or at least some additional citations for follow-up reading) in several areas. Here are a few examples for which I would like to see some additional elaboration:
    1. At the beginning of the introduction, explicitly define the term “relative risk” and discuss why it is a useful metric for clinical and epidemiological studies. Comparison to the definition and interpretation for “odds ratio” would also be helpful to motivate the study.

ANSWER: We have now expanded the beginning of the introduction by adding “The RR is defined as the ratio between the outcome probabilities obtained in two groups, and thus has a direct probabilistic interpretation compared with the more abstract definition of the OR. Hence, while translating an RR of 2 to the statement "The risk in the intervention group is twice as high as in the control group." a similar interpretation would be incorrect for an OR.”

  1. Sections 2.2.4 and 2.3.2 on binomial regression are very brief and could benefit from additional explanation or some citations for further reading. As written, these sections will be very difficult to understand by anyone without a strong statistical background.  In particular, I do not see that 2.3.2 gives the reader sufficient information to replicate the results of this study on their own.

ANSWER: We have now expanded on this section by adding “as binomial regression generally is the preferred strategy to obtain RR estimates from regression models” and referring to “19. Greenland, S. Model-based estimation of relative risks and other epidemiologic measures in studies of common outcomes and in

case-control studies. Am J Epidemiol 2004, 160, 301–305.”

  1. Authors should discuss the difference between frequentist confidence intervals and Bayesian credible intervals. In particular, usage of Bayesian methods is generally not as widespread as frequentist methods, and an explicit definition for a credible interval (and discussion of how it differs in interpretation from a frequentist confidence interval) should be provided.

ANSWER: We have now extended on the interpretation of Bayesian methods in the discussion by adding “Moreover, while Bayesian approaches have an explicit probabilistic interpretation, as a $95\%$ CrI indicates an interval, which with $95\%$ probability contains the true population parameter, conditional on the chosen prior distributions, Bayesian approaches are still not widespread in epidemiological literature [Rietbergen2017], which limits dissemination of their results.” Including a reference to the paper

“20. Rietbergen, C.; Debray, T.P.A.; Klugkist, I.; Janssen, K.J.M.; Moons, K.G.M. Reporting of Bayesian analysis in epidemiologic research should become more transparent. J Clin Epidemiol 2017, 86, 51–58.”

  • Use of the “adding 1” or “moving 1” methods should be treated with extreme caution, as they bring to mind issues of data manipulation/manufacturing.  Wherever possible, it should be clarified that:
    1. Increasing the sample size through the “adding 1” method will lead to narrower confidence intervals, giving an inflated sense of precision.

ANSWER: We have now added the statement “resulting in too narrow confidence intervals” to the corresponding section.

  1. Use of the “moving 1” method will always lead to a more “conservative” estimate of the RR in the sense that it will indicate a weaker association than is actually present.

ANSWER: Thank you for the suggestion. We have made this point more explicit by adding the sentence: “However, it should be noted that this will always lead to a more conservative estimate of the RR in the sense that it will indicate a weaker association than is actually present.”

  1. The authors did a nice job of stating that these methods are not robust for small sample sizes in the discussion; I’d like to see a sentence to this effect added up front in the abstract as well.

ANSWER: Thank you for reminding us of this important point. We have now added the following sentence to the abstract: “However, it is important to note, that the obtained intervals are sensitive to the method chosen, in case of small sample sizes.” (not highlighted in the tracked changes version, due to limitations in the abstract template)

  • A minor issue: please indicate that the use of z* = 1.96 in the confidence intervals in Sections 2.2.1 and 2.2.3 corresponds to a 95% CI; else, replace with the more general z* notation and indicate how these critical values may change to meet whatever confidence level you choose to impose.

ANSWER: Thank you for the relevant observation, we have added the sentence “Here 1.96 is the 0.975 quantile of the normal distribution, corresponding to a 95% confidence interval.” To the first use of this constant.
